# Token-Efficient Long-Term Interest Sketching and Internalized Reasoning for LLM-based Recommendation

**Zhihao Ding**[1,2]  **Jinming Li**[2]  **Shuai Mu**[2*]  **Jieming Shi**[1*]
[1]The Hong Kong Polytechnic University    [2]ByteDance
`huangyuzhang@bytedance.com`, `jieming.shi@polyu.edu.hk`

## Abstract

Large language models (LLMs) can solve complex real-world tasks when prompted to generate chain-of-thought (CoT) reasoning, motivating their use for preference reasoning in recommender systems. However, applying LLM reasoning on recommendation faces two practical challenges. First, LLMs struggle to reason over long, noisy user histories that often span hundreds of items while truncation discards signals needed to capture long-term interests. Second, in decoder-only architectures, CoT requires generating rationale tokens autoregressively, leading to prohibitive inference latency for real-world deployment. To address the challenges, we propose SIREN, a framework that enables effective LLM-based rating prediction via long-term interest sketching and internalized reasoning. First, instead of prompting raw histories, we build a compact, token-bounded interest sketch that preserves persistent preferences and suppresses noise. Specifically, we encode and cluster item descriptions to discover semantic topics, then compress each user's history into a short list of liked and disliked topics, facilitating LLM reasoning. Second, we develop an internalized reasoning strategy for efficient inference. We adopt a two-stage training paradigm: (i) train the LLM to reason explicitly for rating prediction with rule-based reinforcement learning, since ground-truth CoTs are unavailable in recommendation; and (ii) learn to internalize CoT into model parameters through hidden alignment. At inference, the LLM directly generates the rating with near-CoT quality. Extensive experiments show that SIREN reduces average input tokens by $48.7\%$ compared to raw-history prompting, outperforms existing methods while delivering over $100\times$ lower inference latency than CoT-based LLM recommenders. Code and data are available at `https://github.com/TommyDzh/SIREN`.

## 1 Introduction

Large language models (LLMs) have recently demonstrated strong problem-solving abilities, especially when their reasoning capacity is activated by chain-of-thought (CoT) prompting (Wei et al., 2022; Achiam et al., 2023). Reinforcement learning further amplifies this ability, as seen in OpenAI-o1 (Jaech et al., 2024) and DeepSeek-R1 (Guo et al., 2025), enabling LLMs to solve Olympiad-level mathematics (Castelvecchi, 2025) and real-world coding tasks (Jimenez et al., 2024). Motivated by these advances, recent work explores LLMs in recommendation tasks such as user rating prediction (Tsai et al., 2024a; Kim et al., 2025) and next item prediction (Bao et al., 2024; 2025). In rating prediction, for example, an LLM is prompted with a user's behavior history together with a candidate item's description, and is instructed to infer the user's preference and output the rating the user would give. Unlike traditional ID-centric recommenders (Cheng et al., 2016; Zou et al., 2023), LLM-based approaches can exploit rich item semantics, alleviating cold-start, improving generalization, and offering interpretable recommendations (Wang et al., 2024b).

Despite this promise, two practical challenges limit deployment of LLMs in real-world recommenders. First, **long and noisy histories** undermine LLM reasoning for recommendation. In practice, users can have hundreds of interactions within days, producing histories that are lengthy, redundant, and

---

*Corresponding author

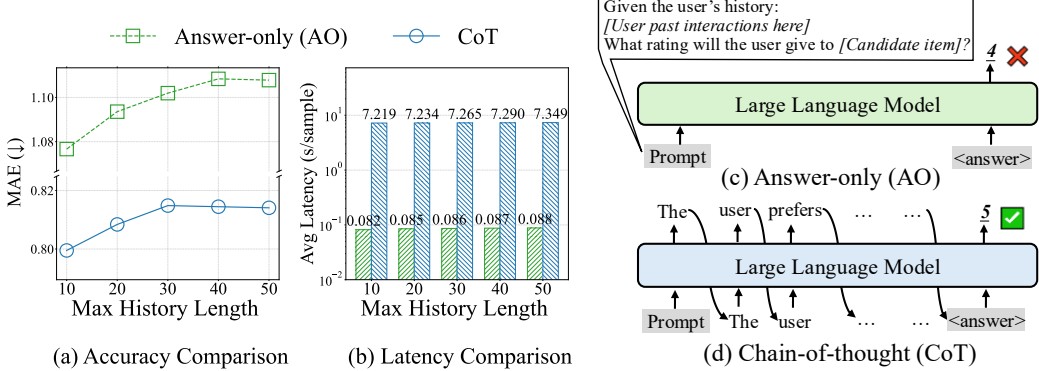

Figure 1: We vary max history length from 10 to 50 and compare answer-only vs CoT of Qwen3-4B in rating prediction on the Movies category of the AmazonReviews dataset. Full setup in Sec. B.2. (a) MAE ($\downarrow$) vs. history length. (b) Average per-sample latency (s). (c) AO: placing `<answer>` immediately after the query `Prompt` signals the model to generate only the final answer. (d) CoT: the model first generates an multi-step reasoning token-by-token, then outputs the final prediction.

noisy (Wang et al., 2025). Naively feeding these raw histories to an LLM degrades performance as LLMs are short of long-context processing (Wang et al., 2024a; Du et al., 2025) and accumulated noise obscures the preference signal and impairs reasoning. As shown in Fig. 1(a), when using LLM for rating prediction, the prediction error increases as the maximum history increases from 10 to 50 items. Prior approaches truncate to the most recent items (Lyu et al., 2024; Tsai et al., 2024a), sacrificing long-term signals (Chang et al., 2023), or prompt the LLM to summarize profiles (Kim et al., 2025; Wang et al., 2025), which still requires processing long histories and adds computational cost. This calls for a token-efficient user representation that preserves robust long-term signals for LLM reasoning.

Second, **latency from explicit reasoning** hinders deployment of LLMs in recommendation. Currently, there are two strategies for LLM inference in rating prediction: answer-only (AO) and chain-of-thought (CoT). AO inference (Fig. 1(c)) takes the prompt and directly generates the rating, whereas CoT (Fig. 1(d)) emits a multi-step rationale before the final rating, and each additional token requires an extra forward pass. Although CoT improves accuracy, the longer output sequence inflates inference latency and serving cost (Lin et al., 2025), which conflicts with production requirements. As shown in Fig. 1(b), CoT incurs per-sample latency over $100\times$ higher than answer-only. To accelerate inference, prior work either distills knowledge into smaller LLMs (Xu et al., 2025) or employs efficient decoding strategies like speculative decoding (Lin et al., 2025) and latent-reasoning (Zhang et al., 2025). However, these approaches still require generating intermediate reasoning tokens, incurring extra time costs.

This work addresses both challenges with SIREN, a framework for leveraging LLM reasoning in rating prediction. SIREN introduces two key components: (i) *Long-Term Interest Sketching.* Instead of prompting with raw, lengthy histories, we build a compact, token-bounded interest sketch that preserves user persistent preferences and suppresses noise. We encode item descriptions and cluster their embeddings to obtain a fixed set of corpus-level semantic topics; each user's history is then aggregated over these topics into a small list of likes and dislikes. We combine this sketch with user recent histories to capture short-term interests, providing a token-efficient yet informative prompt for LLM reasoning. (ii) *Internalized Reasoning.* We aim for answer-only inference that produces the final rating without emitting rationale tokens while retaining the gains of CoT. To this end, we adopt a two-stage training paradigm: first, since ground-truth CoT rationales are unavailable in recommendation, we train explicit CoT reasoning over the interest sketch via rule-based reinforcement learning; second, we learn to internalize CoT by aligning the answer token's hidden states under answer-only decoding to those induced by CoT. Our hidden alignment transfers the effect of CoT into model parameters, enabling near-CoT quality with answer-only latency. We conduct extensive experiments on two real-world recommendation datasets to evaluate SIREN. Results show that SIREN attains high token efficiency and strong rating-prediction accuracy with low inference latency. On Movies, for example, SIREN reduces average input tokens by $48.7\%$ compared to raw-history prompting, lowers MAE by

$20.1\%$ over the runner-up baseline, and delivers over $100\times$ lower inference latency than CoT-based decoding. In summary, our contributions are as follows:

- We identify two deployment challenges for LLM-based recommendation—long, noisy histories and latency from explicit reasoning—and present SIREN, a unified framework that addresses both.
- We propose Long-Term Interest Sketching, a token-efficient user representation that replaces raw, lengthy histories with a compact, corpus-level topic sketch capturing stable preferences.
- We develop a two-stage training paradigm: (i) learn explicit reasoning for rating prediction without CoT labels via RL, and (ii) internalize this reasoning through hidden-state alignment, enabling answer-only decoding with near-CoT quality.
- Through extensive experiments, SIREN achieves lower rating-prediction error than traditional and LLM-based recommender baselines, while delivering over $100\times$ lower inference latency than CoT-based decoding.

## 2 TASK FORMULATION

Let $\mathcal{U}$ be the set of users and $\mathcal{I}$ the set of items. Each item $i \in \mathcal{I}$ has an associated textual description (e.g., title/metadata) denoted $d(i)$. For a user $u \in \mathcal{U}$, let the chronologically ordered interaction history be $\mathcal{H}_u = \{(i, d(i), r_{ui}) : i \in \mathcal{I}_u\}$, where $r_{ui}$ is the observed rating. Given a candidate item $i \in \mathcal{I}$, the goal of rating prediction is to estimate the rating that user $u$ will give to $\hat{r}_{ui}$ based on user history $\mathcal{H}_u$ and the candidate item description $d(i)$.

When using an LLM for rating prediction, the pair $(\mathcal{H}_u, d(i))$ is first converted into a textual prompt by a prompt function $\Phi(\cdot)$, and then fed to the LLM-based recommender model $\mathcal{M}$ to predict:

$$\hat{r}_{ui} = \mathcal{M}\big(\Phi(\mathcal{H}_u, d(i))\big). \tag{1}$$

## 3 SIREN

Fig. 2 illustrates the overall framework of SIREN. SIREN comprises two main components: (i) *Long-Term Interest Sketching* (Sec. 3.1) compresses long, noisy histories into a compact, token-efficient sketch that preserves stable preferences. (ii) *Internalized Reasoning* (Sec. 3.2) uses a two-stage training strategy to retain CoT gains while enabling answer-only decoding, accelerating inference.

### 3.1 LONG-TERM INTEREST SKETCHING

Given a user's history $\mathcal{H}_u$, directly prompting an LLM with raw interactions is suboptimal as histories can be long and noisy. On the contrary, truncating to recent items discards information critical for long-term preferences. In SIREN, we construct a compact *long-term interest sketch* and prompt the LLM with this sketch instead of the raw history. The constructed sketch comprehensively captures user preference signals under strict token limit. As shown in Fig. 2, we first encode all item descriptions and cluster their embeddings to obtain a shared set of corpus-level semantic topics. Given these topics, a user's history is mapped to topic labels and aggregated to yield a small set of interests, partitioned into likes and dislikes according to the average rating. Finally, we form the prompt by concatenating this sketch with the user's recent interactions and the candidate item description, capturing both long-term and short-term preferences. The LLM takes this prompt to predict the rating. We detail each step below.

**Item Topic Discovery.** We represent each item by a semantic topic rather than its raw description. This lets us aggregate many historical items that share a topic, so a user can be summarized by a small set of topics that highlight core interests and discard irrelevant detail. Because users have histories of highly variable length, discovering topics per user is unstable (especially for short histories) and produces topics that are not comparable across users. Instead, we discover corpus-level topics once over the entire item set $\mathcal{I}$. Specially, we first embed item descriptions as:

$$\mathbf{e}(i) = \mathrm{Enc}\big(d(i)\big) \in \mathbb{R}^{d_e}, \quad i \in \mathcal{I}, \tag{2}$$

where $\mathrm{Enc}(\cdot)$ denotes the text encoder and $\mathbf{e}(i)$ is the embedding of item $i$. We then apply $K$-means over all embeddings to obtain $K$ cluster (topic) centers:

$$\{\boldsymbol{\mu}_k\}_{k=1}^{K} = \mathrm{KMeans}(\{\mathbf{e}(i)|i \in \mathcal{I}\}), \tag{3}$$

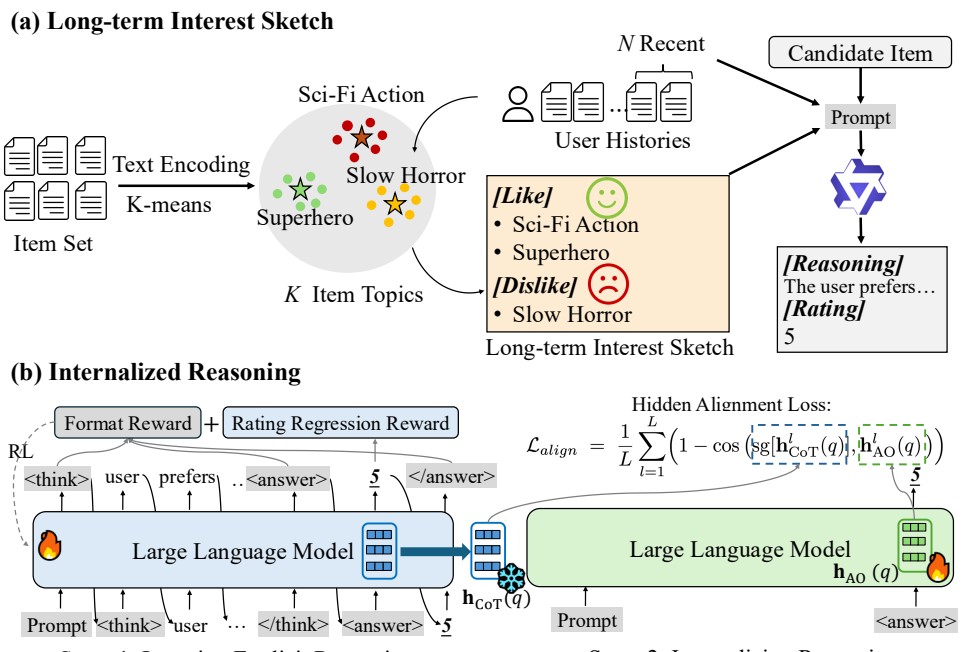

Figure 2: Overview of SIREN. (a) **Long-term interest sketching:** user histories are encoded, clustered into semantic topics, and aggregated into a compact sketch of likes and dislikes, which is combined with recent interactions and the candidate item for LLM-based rating prediction. (b) **Internalized reasoning:** Stage 1 trains explicit CoT reasoning with reinforcement learning (RL); Stage 2 aligns hidden states of answer-only decoding with CoT decoding, enabling answer-only inference with near-CoT quality but much lower latency.

where $\boldsymbol{\mu}_k$ is the $k$-th topic center. Each item is assigned a topic label by finding the nearest center:

$$c_i = \arg \min_{k \in \{1,\dots,K\}} \left\| \mathbf{e}(i) - \boldsymbol{\mu}_k \right\|_2^2. \tag{4}$$

For cluster $k$, we collect the descriptions of the $M$ items closest to its center $\boldsymbol{\mu}_k$ and prompt the LLM to summarize them into a concise textual topic name $\tau_k$.

**User Interest Sketching.** Given item topics $\tau_i \in \{1, \dots, K\}$, we aggregate a user's history $\mathcal{H}_u$ into a compact long-term interest sketch. For the $k$-th topic, the user's interactions within that topic are:

$$\mathcal{H}_u(k) = \{ (i, d(i), r_{ui}) \in \mathcal{H}_u : c_i = k \}, \tag{5}$$

where $\mathcal{H}_u(k)$ is the set of interactions assigned to topic $k$. We partition the topics the user has interacted with into *likes* and *dislikes* based on the average rating per topic:

$$\bar{r}_{u,k} = \frac{1}{|\mathcal{H}_u(k)|} \sum_{(i, r_{ui}) \in \mathcal{H}_u(k)} r_{ui}. \tag{6}$$

Then we threshold the per-topic average and obtain:

$$\mathcal{T}_u^+ = \{ \tau_k : \bar{r}_{u,k} \geq \theta \}, \qquad \mathcal{T}_u^- = \{ \tau_k : 0 < \bar{r}_{u,k} < \theta \}, \tag{7}$$

where $\mathcal{T}_u^+$ and $\mathcal{T}_u^-$ are the sets of topic names the user likes and dislikes, respectively. The user's long-term interest sketch is:

$$\mathcal{S}_u = \left( \mathcal{T}_u^+, \mathcal{T}_u^- \right). \tag{8}$$

This sketch replaces raw, lengthy histories with a small, interpretable topic-level summary that preserves persistent preferences under a strict token budget and suppresses noise, providing a stable user information for LLM reasoning.

**Prompt Construction.** To predict user $u$'s rating for item $i \in \mathcal{I}$, we build the prompt from the user's long-term interest sketch $\mathcal{S}_u$, the $N$ most recent interactions $\mathcal{H}_u^{(N)}$, and the candidate item description $d(i)$. Let function $\Phi(\cdot)$ linearize these components into prompt $\pi_u(i)$:

$$\pi_u(i) = \Phi\left( \mathcal{S}_u, \mathcal{H}_u^{(N)}, d(i) \right). \tag{9}$$

LLM predicts the rating as:

$$\hat{r}_{ui} = \mathcal{M}\big(\pi_u(i)\big) = \mathcal{M}\big(\Phi(\mathcal{S}_u, \mathcal{H}_u^{(N)}, d(i))\big). \tag{10}$$

This construction captures long-term preferences through $\mathcal{S}_u$ and short-term context through $\mathcal{H}_u^{(N)}$ under a strict token budget, providing a concise but informative input for rating prediction.

## 3.2 INTERNALIZED REASONING

While explicit CoT improves rating prediction (Fig. 1), it requires generating many rationale tokens, which inflates inference latency and serving cost. Our goal is answer-only inference that generates only the final rating without losing the quality gains of CoT. As shown in Fig. 2, SIREN adopts a two-stage training paradigm: (i) learn explicit CoT reasoning over the long-term interest sketch prompts for rating prediction; (ii) hidden alignment that trains the CoT model to internalize the explicit reasoning to model parameters. In this way, SIREN enables LLM decoding at inference with near CoT quality. The details are below.

**Learning Explicit Reasoning via Reinforcement Learning.** In the first stage, we train the model to explicitly reason over user sketch prompt $\pi_u(i)$ for rating prediction by generating a rationale followed by a rating $\hat{r}_{ui}$. Since ground-truth CoTs are unavailable in recommendation, we adopt reinforcement learning (RL) to optimize reasoning quality guided by rule-based rewards. We fine-tune LLM with Group Relative Policy Optimization (GRPO) (Shao et al., 2024), a lightweight, critic-free RL update for LLMs.

To adapt GRPO in rating prediction, we combine a *format reward* $s_{\text{format}}$ that enforces the model to generate its reasoning within `<think>`...`</think>` followed by the final prediction within `<answer>`...`</answer>`, and a *rating regression reward* $s_{\text{rate}}$ that maps prediction error into rewards ranges from $[-2, 2]$. Let $r_{ui} \in [a, b]$ denote the ground-truth rating user $u$ has on item $i$, $\hat{r}_{ui} \in [a, b]$ the prediction (clipped to $[a, b]$), and $E_{\max} = b - a$ the maximum possible error. The per-sample reward is defined as:

$$R(\hat{r}_{ui}, r_{ui}) = s_{\text{format}} + \underbrace{\left(2 - \tfrac{4}{E_{\max}}|\hat{r}_{ui} - r_{ui}|\right)}_{s_{\text{rate}}}, \quad s_{\text{format}} = \begin{cases} +1, & \text{if format is correct,} \\ -1, & \text{otherwise.} \end{cases} \tag{11}$$

Here $s_{\text{rate}} \in [-2, 2]$ decreases linearly with the absolute prediction error, and $s_{\text{format}} \in \{-1, +1\}$ enforces the output format. GRPO then optimizes the model using $R(\hat{r}_{ui}, r_{ui})$, encouraging well-formed rationales and ratings close to the ground truth. GRPO algorithm is deferred to Appendix C.

**Learning to Internalize CoT via Hidden Alignment.** In the second stage, we train the LLM to internalize its reasoning. Previously, the model has learned to reason explicitly over long-term interest sketches. The objective of reasoning internalization is to produce the final rating without generating intermediate CoT tokens, which reduces inference latency significantly. To this end, we adopt a hidden-alignment training strategy. We observe that, for a trained LLM $\mathcal{M}_\theta$ from previous stage, the final rating prediction under both answer-only (Fig. 1(c)) and CoT (Fig. 1(d)) decoding depends on the hidden state at the `<answer>` token. Consequently, if we align the answer-only hidden state to its CoT counterpart, the resulting predictions coincide while no CoT tokens are needed.

As shown in Fig. 2, let $\mathbf{h}_{\text{AO}}^l(q)$ denote the hidden state of the `<answer>` query token $q$ at layer $l$ of $\mathcal{M}_\theta$ when only the input prompt is provided, and let $\mathbf{h}_{\text{CoT}}^l(q)$ denote the corresponding hidden state when both the prompt and CoT are present. We fine-tune $\mathcal{M}_\theta$ with the hidden alignment loss:

$$\mathcal{L}_{\text{align}} = \frac{1}{L} \sum_{l=1}^{L} \Big(1 - \cos\big(\text{sg}[\mathbf{h}_{\text{CoT}}^l(q)], \mathbf{h}_{\text{AO}}^l(q)\big)\Big), \tag{12}$$

where $L$ is the number of hidden layers in $\mathcal{M}_\theta$ and $\text{sg}[\cdot]$ denotes the stop-gradient operator. At inference time, even without explicit CoT tokens, the model computes similar hidden states for `<answer>` and achieves rating-prediction quality comparable to explicit reasoning. In practice, we apply low-rank adaptation (LoRA) (Hu et al., 2022) to the key and value projection matrices of each attention layer, following our theoretical motivation presented below.

**Theoretical Justifications.** We analyze why hidden alignment enables internalized reasoning below.

**Theorem 1** (KV Adaptation Equivalence). *Let $q$ denote the answer-token query, $Q$ the set of question prompt tokens, and $Z$ the set of CoT reasoning tokens. For the $l$-th attention layer with key/value projections $(W_K^l, W_V^l)$, define*

$$\mathbf{h}_{\mathrm{AO}}^l(q) \;=\; F^l([Q, q]; [W_K^l, W_V^l]), \qquad \mathbf{h}_{\mathrm{CoT}}^l(q) \;=\; F^l([Q, Z, q]; [W_K^l, W_V^l]),$$

*where $F^l$ returns the layer-$l$ hidden state at $q$ produced by a standard attention block followed by feed-forward network (FFN) block. Define the* updated *answer-only hidden*

$$\tilde{\mathbf{h}}_{\mathrm{AO}}^l(q) \;:=\; F^l([Q, q]; [W_K^l + \Delta W_K^l, \; W_V^l + \Delta W_V^l]).$$

*Then, under a first-order (linearized) treatment of the attention and FFN at layer l, there exist updates $(\Delta W_K^l, \Delta W_V^l)$ such that*

$$\tilde{\mathbf{h}}_{\mathrm{AO}}^l(q) \;=\; \mathbf{h}_{\mathrm{CoT}}^l(q).$$

This provides the motivational basis for our KV-targeted low-rank adaptation strategy for hidden alignment. The full proof is deferred to Appendix A.

## 4 EXPERIMENTS

In this section, we conduct experiments to address the following research questions:

- **RQ1:** How does SIREN perform on rating prediction under compared with classic recommenders and recent LLM-based baselines?
- **RQ2:** What is the inference efficiency of SIREN versus LLM-based baselines?
- **RQ3:** Does our interest sketch improve accuracy over raw history and summary-based methods?
- **RQ4:** Does the internalize CoT via hidden alignment effectively bridge the gap between answer-only and CoT decoding?

### 4.1 EXPERIMENT SETUP

**Datasets and Baselines.** We evaluate on the *Books* and *Movies* categories from the Amazon Reviews 2023 dataset (Hou et al., 2024)[1]. Dataset statistics are reported in Table 1. The Books split follows (Kim et al., 2025); the Movies split is curated by taking the first 150k interactions from the Movies category of Amazon Reviews 2023. Following prior work (Kim et al., 2025; Bismay et al., 2025), we apply the 5-core filter, iteratively removing users and items with fewer than five interactions, and perform stratified sampling to keep positive (ratings $\geq$ 4) and negative (ratings $\leq$ 3) examples relatively balanced (Bismay et al., 2025). We adopt a *leave-last-out* split (Hou et al., 2024): for each user, the last two interactions are held out for validation and test, respectively, and the remainder are used for training.

We compare SIREN with 9 competitors in 3 categories. (i) Dataset Statistics (Kang et al., 2023), including *User Avg. Rating* and *Item Avg. Rating*. (ii) Traditional Recommendation Methods, including *Matrix Factorization (MF)* (Rendle et al., 2009) and *P5* (Geng et al., 2022). (iii) LLM-based Recommendation, including *LLM4Rate* (Kang et al., 2023) (using DeepSeek–R1 and Qwen3–4B as backends), respectively, *Rec-SAVOR* (Tsai et al., 2024b), and *EXP3RT* (Kim et al., 2025). Details of these baselines and their implementations are provided in Appendix B.3.

Table 1: Dataset Statistics

| Dataset | # Train | # Valid | # Test | # User | # Item |
|---------|---------|---------|--------|--------|--------|
| Book | 94075 | 12222 | 11708 | 10440 | 9753 |
| Movie | 22097 | 2629 | 2629 | 2629 | 10874 |

**Implementation Details and Evaluation Metrics.** For SIREN and EXP3RT, we use Qwen3-4B (Yang et al., 2025) as the backend LLM. We use BGE-M3 (Chen et al., 2024a) as the text encoder, set number of corpus-level semantic topics $K = 20$, number of descriptions to generate topic name $M = 100$, threshold for separating likes/dislikes $\theta = 4$. For prompt construction, we include the most recent $N$ user interactions, with $N = 30$ for the Books dataset and $N = 10$ for the Movies dataset. For explicit reasoning, we train with GRPO using the VERL framework (Sheng et al., 2024). Table 15 summarizes the key hyperparameters for each stage of our pipeline. Additional implementation details are in Appendix B. For rating prediction, we report *MAE* (Mean Absolute Error) and *RMSE*

---

[1]https://amazon-reviews-2023.github.io/

Table 2: Overall performance for rating prediction in MAE (↓) and RMSE (↓). Rank indicates the average per-method rank achieved across all datasets and metrics. **Bold**: best. Underline: runner-up.

| Methods | Books | | Movies | | Rank |
|---|---|---|---|---|---|
| | MAE (↓) | RMSE (↓) | MAE (↓) | RMSE (↓) | |
| *Dataset Statistics* | | | | | |
| User Avg. Rating | 0.4538 | 0.8396 | 0.9975 | 1.3419 | 5.25 |
| Candidate Item Avg. Rating | 0.5071 | 0.7567 | 1.5205 | 2.0683 | 7 |
| *Traditional Recommendation Methods* | | | | | |
| MF | 0.5740 | 0.7259 | 1.1604 | 1.3060 | 5.25 |
| P5 | 0.5591 | 0.7939 | 0.8655 | 1.3406 | 4.75 |
| *LLM-based Recommendation* | | | | | |
| LLM4Rate (DeepSeek-R1) | 0.4509 | 0.8329 | 0.9222 | 1.4047 | 4.5 |
| LLM4Rate (Qwen3-4B) | 0.4730 | 0.8916 | 0.7995 | 1.4797 | 5.75 |
| Rec-SAVOR | 0.4893 | 0.8902 | 1.0884 | 1.5183 | 7.25 |
| Exp3RT | 0.4001 | 0.7567 | 0.9521 | 1.4646 | 4 |
| SIREN | **0.3510** | **0.6887** | **0.7603** | **1.2924** | **1** |

(Root Mean Squared Error) following (Chen et al., 2024b; Kim et al., 2025). Lower values indicate better performance for both metrics. Following EXP3RT, we conduct each experiment with three random seeds and report the mean.

## 4.2 OVERALL PERFORMANCE

**Rating Prediction Accuracy (RQ1).** Table 2 reports rating prediction results for SIREN and baselines. We summarize three observations. (1) SIREN achieves the best MAE and RMSE on both datasets, often by a large margin. For example, on Books, SIREN improves over the strongest LLM baseline, EXP3RT, by 12.45% in MAE and 8.99% in RMSE, indicating more accurate predictions with fewer large errors. (2) LLM-based methods generally outperform heuristic/statistical and traditional recommendation baselines in MAE, reflecting the benefit of leveraging item semantics. However, we observe weaker gains in RMSE, sometimes trailing MF. A closer look suggests label-imbalance bias (datasets skewed toward positive ratings) leads LLMs to over-predict high ratings (e.g., ≥ 4), which increases large-error penalties. By contrast, SIREN mitigates this via RL-based CoT training (GRPO) and subsequent internalization, improving robustness on minority low-rating cases. (3) Among LLM baselines, no single prior method dominates. Notably, EXP3RT, fine-tuned on CoT labels, does not consistently surpass LLM4Rate with zero-shot prompting, highlighting the importance of both user representation and training strategy. In contrast, SIREN 's consistent improvements stem from long-term interest sketching, which preserves comprehensive, noise-resilient preference signals under a strict token budget, and internalized reasoning, which retains CoT-quality prediction while decoding answer-only.

**Inference Efficiency (RQ2).** We compare average generated output tokens and per-sample inference latency (s) for SIREN with answer-only decoding (SIREN-AO), its explicit CoT variant from Stage 1 Sec. 3.2 (SIREN-CoT), and LLM-based competitors (EXP3RT, LLM4Rate). We use vLLM (Kwon et al., 2023) on a single H20 GPU with batch size 1. We set maximum output size as 1024 tokens and report per-sample inference latency (seconds) and the average number of generated output tokens, averaged over the full test set of each dataset. As shown in Fig. 3: (i) explicit reasoning emits hundreds of rationale tokens, whereas SIREN-AO generates a single answer token; (ii) accordingly, SIREN-AO incurs substantially lower latency (about 0.013 s/sample on both datasets), yield-

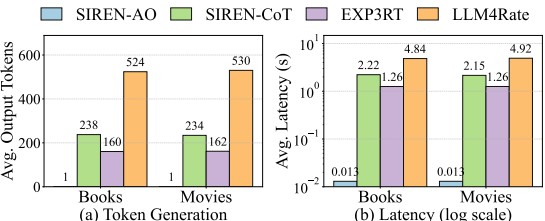

Figure 3: Comparison of LLM-based methods in average latency (s/sample) versus average number of generated output tokens (per sample).

ing over 100× speedup. Together with the accuracy results in Table 2, these findings indicate that SIREN achieves state-of-the-art effectiveness and better efficiency for rating prediction.

## 4.3 IN-DEPTH ANALYSIS

**Study on Long-term Interests Sketching (RQ3).** To examine the effectiveness of our proposed long-term interest sketch (Sec. 3.1), we compare different user modeling strategies for rating prediction. We fine-tune the LLM with answer-only labels, and evaluate prompts constructed from: (i) the most recent $N$ interactions (*Recent History*), (ii) recent history augmented with our proposed sketch (*+Sketch*), (iii) recent history extended with additional past interactions until reaching the token budget (*+More History*), and (iv) user profiles summarized by LLMs as in Kim et al. (2025) (*+Profile*).

Table 3 shows that our sketch achieves consistent improvements over recent history, attaining the best performance except for one MAE case on Books where it ranks second. Adding more histories sometimes reduces MAE but degrades RMSE, suggesting that longer raw histories introduce noise that amplifies large errors. In contrast, LLM-generated profiles fail to improve upon recent history, highlighting the difficulty of off-the-shelf summarization in capturing key user preferences for accurate rating prediction.

Table 3: Comparison of performance under different user modeling designs.

| Strategy | Books | | Movies | |
|---|---|---|---|---|
| | MAE | RMSE | MAE | RMSE |
| Recent History | 0.3547 | 0.7131 | 0.7775 | 1.3635 |
| +Sketch (ours) | 0.3536 | **0.7114** | **0.7695** | **1.3556** |
| +More history | **0.3535** | 0.7153 | **0.7695** | 1.3723 |
| +Profile | 0.3563 | 0.7244 | 0.7779 | 1.3872 |

**Study on Internalized Reasoning (RQ4).** We evaluate strategies for converting a GRPO-trained, explicit CoT model into answer-only (AO) inference, with all variants initialized from the same Stage-1 (GRPO-CoT) checkpoint. We post-train GRPO-trained model with: (i) *CE*: supervised fine-tuning on rating labels using cross-entropy loss; (ii) *KD*: logits-based knowledge distillation from the GRPO-CoT teacher (Hinton, 2014); (iii) *KD+CE*: joint distillation and cross-entropy; (iv) *HA*: our hidden alignment loss; (v) *HA+CE*: hidden alignment combined with cross-entropy. Fig. 4 summarizes results, with the GRPO-CoT reference shown as a red dashed line. Across Books and Movies, HA consistently yields the lowest error and closely tracks the GRPO-CoT teacher,

indicating that aligning the hidden states effectively internalizes CoT into the model parameters while preserving accuracy under answer-only decoding. Interestingly, HA even surpasses the GRPO-CoT teacher in Books, likely because aligning hidden states transfers the rationale structure while avoiding the noise and variance of explicitly generating CoT tokens. In contrast, *HA+CE* underperforms *HA*, suggesting token-level CE pulls the model toward label-only fitting which conflicts with latent structure induced by CoT. KD-based variants remain behind HA, showing the benefit of aligning latent states over only matching output distributions.

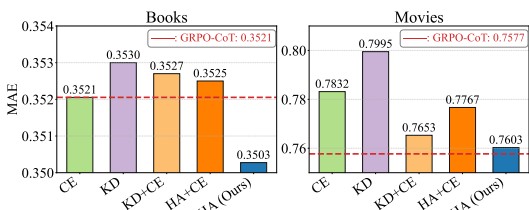

Figure 4: Comparison of strategies for internalizing CoT reasoning, All variants are initialized from the Stage-1 GRPO-CoT model; the GRPO-CoT teacher is shown as a red dotted line.

**Effect of Different LoRA Target Modules.** To internalize CoT into the model parameters, we apply LoRA adapters and train with the hidden-alignment loss in Eq. 12. Table 4 compares SIREN when adapting different target module sets: (i) *All-linear* (all linear projections), (ii) *QKV*

(all attention projections), (iii) *QV* (query and value), and (iv) *FFN* (feed-forward layers only). Overall, adapting KV yields the best results on almost all datasets and metrics, while increasing the adaptation scope to all-linear does not consistently improve accuracy under the same training recipe. These findings align with Theorem 1, which motivates KV as a sufficient locus to absorb CoT effects, even though broader adaptation can sometimes be competitive.

Table 4: Different LoRA target modules.

| Modules | Books | | Movies | |
|---|---|---|---|---|
| | MAE | RMSE | MAE | RMSE |
| KV | **0.3503** | **0.6949** | 0.7581 | **1.2947** |
| all-linear | 0.3505 | 0.6992 | **0.7577** | 1.2987 |
| QKV | 0.3505 | 0.6977 | 0.7581 | 1.2985 |
| QV | 0.3508 | 0.6971 | 0.7623 | 1.2851 |
| FFN | 0.3504 | 0.6979 | 0.7604 | 1.3073 |

**Ablation Study.** We conduct an ablation study that progressively incorporates our proposed designs. In Table 5, we begin with answer-only fine-tune LLM on the most recent history (*AO-SFT (Recent)*). Adding our proposed long-term interest sketch in Sec. 3.1(*AO-SFT (+Sketch)*) consistently reduces MAE/RMSE while keeping latency nearly unchanged, confirming the benefit of token-efficient user

Table 5: Ablation Study.

| Method | Books | | | Movies | | |
|---|---|---|---|---|---|---|
| | MAE (↓) | RMSE (↓) | Latency (↓) | MAE (↓) | RMSE (↓) | Latency (↓) |
| AO–SFT (Recent) | 0.3547 | 0.7131 | **0.011** | 0.7775 | 1.3635 | **0.012** |
| AO–SFT (+Sketch) | 0.3536 | 0.7114 | 0.013 | 0.7695 | 1.3556 | 0.013 |
| CoT–GRPO (+Sketch) | 0.3521 | 0.6977 | 2.22 | **0.7577** | **1.2821** | 2.15 |
| AO–HA (from C) | **0.3503** | **0.6887** | 0.013 | 0.7603 | 1.2924 | 0.013 |

Table 6: Full item descriptions vs. brief text (title+category).

| Method | Books | | Movies | |
|---|---|---|---|---|
| | MAE | RMSE | MAE | RMSE |
| Sketch (detail) | **0.3536** | **0.7114** | **0.7695** | 1.3556 |
| Sketch (brief) | 0.3594 | 0.7159 | 0.7706 | 1.3550 |

representation. Replacing answer-only supervision with GRPO training (Sec. 3.2) that enables explicit CoT reasoning (*GRPO-CoT (+Sketch)*) improves MAE but incurs more than $200\times$ higher latency due to autoregressive rationale generation. Finally, applying our hidden alignment (*AO-HA (+Sketch)*) internalizes the reasoning ability into model parameters, achieving the best MAE with answer-only latency. These highlight that long-term interest sketching improves representation, GRPO strengthens reasoning, and hidden alignment successfully bridges CoT accuracy with AO efficiency.

**Comparison of Full Item Descriptions vs. Brief Text.** We assess the robustness of long-term interest sketching when item text is limited. We construct a brief variant using only titles and categories (excluding detailed descriptions) for item embeddings and topic clustering, and fine-tune with answer-only supervision. Compared to the main setting (full item descriptions), Table 6 shows that brief text yields a small but consistent drop in performance, confirming the value of richer textual signals for topic discovery and sketch quality. However, the degradation is minor, indicating that pretrained LLMs can extract useful semantics even from short texts. Since full descriptions are available in our datasets (Hou et al., 2024) and typically accessible in practice, we recommend using richer item text for encoding and clustering when possible.

**Ablation on Rating Reward.** To assess the impact of the rating reward, we ablate the linear, distance-aware reward $s_{\text{rate}}$ (Eq. 11) with a binary exact-match reward, keeping the format reward and all other GRPO settings unchanged. Specifically, the binary variant assigns $+2$ for

Table 7: Ablation on the rating reward.

| Method | Books | | Movies | |
|---|---|---|---|---|
| | MAE | RMSE | MAE | RMSE |
| Binary exact-match | 0.3555 | 0.6997 | 0.8094 | 1.4201 |
| Regression ($s_{\text{rate}}$) | **0.3520** | **0.6962** | **0.7577** | **1.2822** |

an exact rating match and $-2$ otherwise; the format reward and GRPO recipe are unchanged. Table 7 shows that the distance-aware $s_{\text{rate}}$ consistently outperforms the binary reward across datasets and metrics. This improvement is due to (i) denser learning signals—near-correct predictions receive graded credit rather than all-or-nothing feedback—and (ii) smoother updates, as the reward varies continuously with error, yielding more stable GRPO training.

**Different Backbones.** To assess SIREN across LLM backbones and scales, we substitute Qwen3-4B with *Llama-3.2-3B-Instruct* under identical training settings. We report both the explicit CoT variant trained with GRPO (SIREN–CoT) and the internalized reasoning version (SIREN–AO, 1 token), alongside Exp3RT on each backbone. As shown in Table 8,

Table 8: Different Backbones.

| Backbone | Method | Books | | Movies | |
|---|---|---|---|---|---|
| | | MAE ↓ | RMSE ↓ | MAE ↓ | RMSE ↓ |
| Qwen3-4B | Exp3RT | 0.4001 | 0.7567 | 0.9521 | 1.4646 |
| | SIREN–CoT | 0.3521 | 0.6962 | **0.7577** | **1.2822** |
| | SIREN–AO | **0.3503** | **0.6887** | 0.7603 | 1.2924 |
| Llama-3.2-3B | Exp3RT | 0.3616 | 0.7048 | 0.9209 | 1.4157 |
| | SIREN–CoT | **0.3481** | **0.6956** | **0.7524** | **1.2682** |
| | SIREN–AO | 0.3534 | 0.7042 | 0.7622 | 1.2902 |

SIREN–CoT on Llama-3.2-3B slightly outperforms Qwen3-4B, while SIREN–AO achieves similar results and maintains the 1-token latency. On both backbones, SIREN consistently surpasses Exp3RT, demonstrating effectiveness and robustness across architectures and scales.

**Candidate Reranking on IMDB.** To assess generalizability beyond rating prediction, we follow the EXP3RT protocol and evaluate SIREN on candidate reranking using the IMDB dataset (Kim et al., 2024). LightGCN (He et al., 2020) retrieves the

Table 9: IMDB candidate reranking. Metrics are Recall/nDCG (↑); latency per query in seconds (↓).

| Method | R@3 | R@5 | nDCG@3 | nDCG@5 | Latency (s) |
|---|---|---|---|---|---|
| EXP3RT | 0.0161 | 0.0578 | 0.0149 | 0.0413 | 1.56 |
| SIREN | **0.0256** | **0.0663** | **0.0256** | **0.0426** | **0.03** |

top-20 candidates per user; LLM-based models rerank these by predicted relevance under matched token budgets and identical candidate lists. We compare SIREN with EXP3RT. As shown in Table 9, SIREN consistently outperforms EXP3RT across all metrics, e.g., SIREN improves Recall@5 by 14.7% and achieves $52\times$ lower latency, demonstrating higher efficacy and efficiency. Combined with our rating-prediction results, these findings support SIREN 's effectiveness and generalizability.

**More Experiments.** In Appendix B.4, we evaluate the influence of number of topics $K$ in Fig. 5, effect of different hidden alignment distance function in Table 10, and qualitative analysis on GRPO-trained reasoning over the interest sketch in Table 14.

## 5 RELATED WORK

**LLMs-based Recommendation.** Inspired by the success of LLMs across multiple domains (Feng et al., 2023; Shao et al., 2024; Feng et al., 2024), researchers increasingly explore LLMs as end-to-end recommenders for rating prediction (Tsai et al., 2024a; Kim et al., 2025), sequential recommendation (Chen et al., 2024b; Zheng et al., 2024), and next-item prediction (Bao et al., 2024; 2025). Early work evaluates LLMs for rating prediction via prompt engineering (Achiam et al., 2023; Grattafiori et al., 2024), reporting promising performance, especially after fine-tuning (Kang et al., 2023), along with improved explainability and generalization (Liu et al., 2023). To further enhance preference modeling, recent studies leverage LLM reasoning for recommendation (Tsai et al., 2024b; Kim et al., 2025; Bismay et al., 2025). Rec-SAVOR (Tsai et al., 2024b) and ReasoningRec (Bismay et al., 2025) employ a stronger teacher LLM to generate rationales for user preferences, which are then used to fine-tune a smaller student model, improving recommendation accuracy and providing human-interpretable explanations. To address long-term user preferences beyond truncated histories, several works incorporate long-range signals into LLM-based recommenders. EXP3RT (Kim et al., 2025) employs a teacher LLM to generate user/item profiles and step-by-step reasoning, which supervise student fine-tuning. LLM-TUP (Sabouri et al., 2025) uses LLMs to produce short-term and long-term preference descriptions, which are encoded and fused via attention for richer user embeddings. HyMiRec (Zhou et al., 2025) reconstructs full historical sequences using residual codebook quantization, then aggregates them with a lightweight recommender into coarse interest embeddings. Unlike prior work, SIREN targets two key challenges in LLM-based rating prediction. First, instead of truncating histories (Tsai et al., 2024b; Bismay et al., 2025) or relying on free-form summaries (Kim et al., 2025; Sabouri et al., 2025), we introduce a token-budgeted interest sketch that discovers dataset-level topics once and represents each user with a small, human-readable like/dislike list, stable under strict token budgets. Second, to mitigate the latency of explicit reasoning, we train the model to internalize CoT, enabling answer-only inference that achieves near-CoT quality with substantially lower latency.

**Inference Acceleration for LLM-based Recommendation.** Despite strong accuracy, LLMs face deployment challenges in recommendation due to the latency of autoregressive decoding (Lin et al., 2025; Xu et al., 2025). A common direction is knowledge distillation (KD), transferring knowledge from a large teacher to a smaller student to reduce parameters and speed up inference (Wang et al., 2024c; Xu et al., 2025). More recently, efficient reasoning strategies have been explored in recommendation. For example, AtSpeed (Lin et al., 2025) incorporates speculative decoding to accelerate generative recommendation by drafting multiple top-K item sequences with a lightweight model and verifying them through a target LLM, optimizing both top-K alignment and verification efficiency. LatentR3 (Zhang et al., 2025) introduces a latent-reasoning module that generates a small number of autoregressive latent tokens instead of long text tokens and trains them with a modified GRPO framework. However, these approaches still rely on producing intermediate tokens, which introduces extra decoding steps. In contrast, SIREN eliminates rationale generation entirely by internalizing CoT into model parameters, enabling answer-only decoding while retaining the quality benefits of explicit reasoning.

## 6 CONCLUSION

We studied LLM-based rating prediction and addressed two practical challenges: long, noisy histories hinder an LLM's ability to reason about user preferences, and explicit CoT decoding incurs prohibitive inference latency. We introduced SIREN, a framework that addresses both issues. First, our proposed token-efficient long-term interest sketching compresses a user's history into a compact sketch that preserves persistent preferences while suppressing noise. Second, to enable efficient inference, we internalize reasoning via a two-stage procedure: the model is trained to reason explicitly with reinforcement learning, then aligned to produce answer-only outputs by transferring CoT effects into the parameters through hidden alignment. Extensive experiments show that SIREN improves accuracy under strict token budgets and achieves near-CoT quality at answer-only latency.

ACKNOWLEDGMENTS

This work is supported by grants from the Research Grants Council of Hong Kong Special Administrative Region, China (No. PolyU 15205224), and Smart Cities Research Institute (SCRI) P0051036-P0050643.

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

# A PROOF OF THEOREM 1

*Proof.* Consistent with previous works (Li et al., 2024), we omit the softmax operation and the scaling factor to approximate standard attention as relaxed linear attention for qualitative analysis. We instantiate the layer-$l$ map by the linear attention form

$$F^l([X]; [W_K^l, W_V^l]) \;=\; (XW_V^l)(XW_K^l)^\top Q_q^l,$$

where $X$ is the token matrix at the layer input and $Q_q^l$ is the layer-$l$ query for token $q$.

**CoT/AO difference reduces to a single additive term.** Let $X_Q$ and $X_Z$ be the embeddings of the question tokens $Q$ and the CoT tokens $Z$, respectively. Then

$$\mathbf{h}_{\mathrm{AO}}^l(q) = (X_Q W_V^l)(X_Q W_K^l)^\top Q_q^l,$$
$$\mathbf{h}_{\mathrm{CoT}}^l(q) = ([X_Q; X_Z]W_V^l)([X_Q; X_Z]W_K^l)^\top Q_q^l$$
$$= (X_Q W_V^l)(X_Q W_K^l)^\top Q_q^l \;+\; (X_Z W_V^l)(X_Z W_K^l)^\top Q_q^l.$$

Hence the residual is

$$\Delta^\star(q) \;:=\; \mathbf{h}_{\mathrm{CoT}}^l(q) - \mathbf{h}_{\mathrm{AO}}^l(q) \;=\; (X_Z W_V^l)(X_Z W_K^l)^\top Q_q^l.$$

**Exact equality via span containment.** Consider answer-only with KV updates:

$$\tilde{\mathbf{h}}_{\mathrm{AO}}^l(q) = \big(X_Q(W_V^l + \Delta W_V^l)\big)\big(X_Q(W_K^l + \Delta W_K^l)\big)^\top Q_q^l.$$

Expanding to first order in $(\Delta W_V^l, \Delta W_K^l)$, the equality $\tilde{\mathbf{h}}_{\mathrm{AO}}^l(q) = \mathbf{h}_{\mathrm{CoT}}^l(q)$ requires

$$X_Q \Delta W_V^l (X_Q W_K^l)^\top Q_q^l \;+\; X_Q W_V^l (X_Q \Delta W_K^l)^\top Q_q^l \;=\; (X_Z W_V^l)(X_Z W_K^l)^\top Q_q^l.$$

Define $a := (X_Q W_K^l)^\top Q_q^l$ and $b := X_Q^\top Q_q^l$. In the nondegenerate case $a \neq 0$ or $b \neq 0$, the set of attainable left-hand-side directions contains $\mathrm{Col}(X_Q)$. Consequently, the linear system is solvable iff

$$(X_Z W_V^l)(X_Z W_K^l)^\top Q_q^l \;\in\; \mathrm{Col}(X_Q).$$

The stronger span containments

$$\mathrm{Col}(X_Z W_V^l) \subseteq \mathrm{Col}(X_Q W_V^l) \quad \text{and} \quad \mathrm{Col}(X_Z W_K^l) \subseteq \mathrm{Col}(X_Q W_K^l)$$

are sufficient for solvability but not necessary.

**General case (least-squares projection).** When the span conditions fail, the least-squares problem

$$\min_{\Delta W_K^l, \Delta W_V^l} \left\| X_Q \Delta W_V^l (X_Q W_K^l)^\top Q_q^l + X_Q W_V^l (X_Q \Delta W_K^l)^\top Q_q^l - \Delta^\star(q) \right\|_2^2$$

has a solution that equals the orthogonal projection of $\Delta^\star(q)$ onto the subspace reachable by KV perturbations. This yields the claimed best (unconstrained) approximation.

$\square$

# B EXPERIMENT

## B.1 IMPLEMENTATION DETAILS

Table 15 summarizes the key hyperparameters for each stage of our pipeline. Additional specifics are provided below.

**GRPO Training.** For GRPO training in Sec. 3.2, we use the VERL framework (Sheng et al., 2024). The configuration includes `max_prompt_length=2046`, `max_response_length=512`, constant learning rate $1 \times 10^{-6}$, KL loss coefficient 0.001, micro-batch size per GPU 32, rollout number 8, and training for 2 epochs.

**Hidden Alignment Training.** For hidden alignment in Sec. 3.2, we build on TRL[2]. We fine-tune the model with parameter-efficient fine-tuning via LoRA, applied to the key and value projections of each attention layer. LoRA hyperparameters are set to $r = 8$, `lora_alpha`=16, and `lora_dropout`=0.05. Training uses DeepSpeed ZeRO Stage-2 for memory-efficient distributed optimization. We adopt AdamW with learning rate $1 \times 10^{-4}$, constant schedule with 100 warmup steps, and train for 3 epochs with batch size 64 on $8 \times$ H20 GPUs.

**Latency Evaluation.** For the latency study in Fig. 3, we use vLLM (Kwon et al., 2023) on a single H20 GPU with batch size 1. We set `max_response_length` as 1024 tokens and report per-sample inference latency (seconds) and the average number of generated output tokens, averaged over the full test set of each dataset.

## B.2 MOTIVATIONAL EXPERIMENT SETUP

For the introduction experiment, we evaluate Qwen3–4B without post-training on the *Books* split. We adopt the LLM4Rate prompt (Kang et al., 2023) to instruct the model to perform rating prediction from user history and candidate metadata, and measure performance while varying the maximum history length.

## B.3 BASELINES

We provide more descriptions and implementation details of the baselines in Sec. 4.

- **MF** (Rendle et al., 2009). A representative latent–factor collaborative filtering method for rating prediction based solely on user–item interactions. We use the Surprise Python library[3] for implementation and tune standard hyperparameters on the validation set.

- **P5** (Geng et al., 2022). A foundation model for recommendation. P5 unifies diverse recommendation tasks by converting user–item interactions, item metadata, and user reviews into natural language sequences and training with a language modeling objective. We adopt the official implementation[4] and fine-tune a T5 backbone for the rating-prediction task using the authors' prompt format.

- **LLM4Rate** (Kang et al., 2023). Prompt-based rating prediction with off-the-shelf LLMs. The prompt includes the user's recent history (item titles, genres, ratings) and the candidate item's title and genres. In our setup, we evaluate DeepSeek-R1 and Qwen3-4B without fine-tuning, instructing the model to generate a rationale followed by a rating. The number of recent interactions follows our main setting ($N = 30$ for Books, $N = 10$ for Movies).

- **Rec-SAVOR** (Tsai et al., 2024b). Uses an off-the-shelf LLM to produce a rationale and final rating from a compact window of recent interactions. Following the original design, we use at most 10 recent items and include rich item fields (title, description, categories, price) together with the corresponding user review. We adopt the authors' prompt template and use DeepSeek-R1 as the backbone LLM.

- **EXP3RT** (Kim et al., 2025). Leverages user reviews to build profiles and distill reasoning. A teacher LLM first extracts preference signals from each review and summarizes user and item profiles; it then generates step-by-step rationales and target ratings conditioned on these profiles. The resulting pairs supervise a smaller student LLM. We use the official code release[5]. Following their setup, we use DeepSeek-R1 to generate preferences/profiles/rationales and fine-tune Qwen3-4B as the student, matching our backbone in SIREN. Because their profiles include average-rating statistics, we compute these statistics on the training split only to avoid leakage (the released averages differ from train-only values); we report results using the recomputed statistics.

## B.4 MORE EXPERIMENTAL RESULTS

**Influence of the number of topics $K$.** We study how the number of corpus-level topics ($K$) in the interest sketch affects performance. As shown in Fig. 5, we vary $K \in \{10, 20, 30, 40, 50\}$ on Books

---

[2] https://huggingface.co/docs/trl/v0.19.1/en/sft_trainer
[3] https://surpriselib.com/
[4] https://github.com/jeykigung/P5
[5] https://github.com/jieyong99/EXP3RT

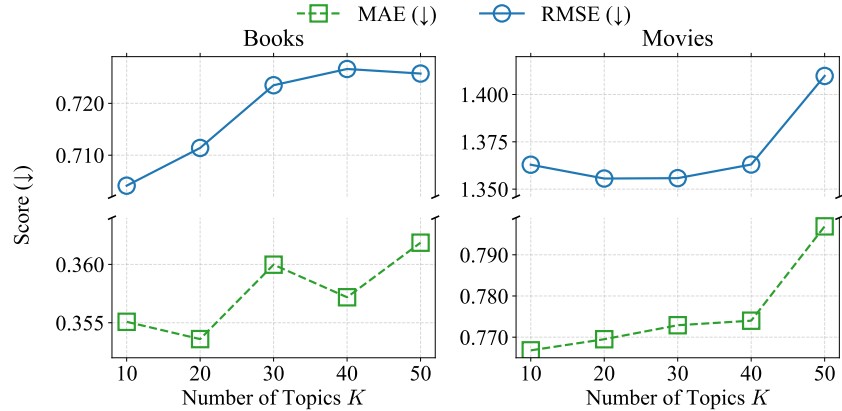

Figure 5: Performance of SIREN when varying number of topics $K$.

Table 10: Performance of loss function used in Eq. 12.

| Method | Books | | Movies | |
|---|---|---|---|---|
| | MAE | RMSE | MAE | RMSE |
| answer-only SFT | 0.3527 | 0.7214 | 0.7695 | 1.3556 |
| GRPO | 0.3521 | 0.6962 | 0.7577 | 1.2822 |
| Cosine | **0.3503** | **0.6930** | **0.7603** | **1.2924** |
| MAE | 0.3523 | 0.6937 | 0.7687 | 1.3001 |
| MSE | 0.3528 | 0.6940 | 0.7679 | 1.2939 |

and Movies and report MAE/RMSE. SIREN performs best with smaller $K$ (e.g., 10–20). Because each dataset is already category-homogeneous (books or movies), large $K$ fragments items into overly fine-grained topics that capture trivial distinctions, weakening preference modeling and risking overfitting. Balancing accuracy and stability across datasets, we set $K! =! 20$ for the main experiments to avoid excessive hyperparameter search while remaining within the empirically favorable range.

**Effect of Different Hidden Alignment Distance Function.** In the hidden alignment loss Eq. 12, we default to $1 - \cos$ similarity to measure the distance between hidden states of CoT and AO. Here, we evaluate alternative distance measures, including mean absolute error (MAE) and mean squared error (MSE). Results are reported in Table 10, with AO-SFT and GRPO-CoT included for reference. We observe that cosine consistently achieves the best performance, suggesting that preserving directional information is more important than matching absolute magnitudes when aligning hidden states.

**Training Cost Comparison.** We compare SIREN against EXP3RT under the same hardware ($8 \times$H20). Because SIREN trains in two stages, we report Stage-1 GRPO and Stage-2 hidden alignment (HA) separately, as well as the total wall-clock time. As shown in Table 11, SIREN's cost is dominated by Stage-1 GRPO, while Stage-2 HA is lightweight. Despite the higher training time, SIREN delivers substantially lower inference latency (Fig. 3) and better accuracy (Table 2). To further reduce cost, we plan to explore coreset/importance data selection (Jiang et al., 2025) and dynamic sampling for RL (Yu et al., 2025).

## C    GRPO DETAILS

Group Relative Policy Optimization (GRPO) (Shao et al., 2024) is a reinforcement learning algorithm designed to activate LLM reasoning without relying on ground-truth chain-of-thought (CoT) labels. Unlike RL algorithms like PPO (Schulman et al., 2017), which estimates the advantage using a learned value function, GRPO computes advantages in a group-relative manner. Specifically, for each input $(q, a)$, the old policy $\pi_{\theta_{old}}$ samples a group of $G$ candidate responses $\{o_i\}_{i=1}^{G}$. Each response receives a rule-based reward $R_i$ (e.g., correctness of the final rating). The normalized group-relative

Table 11: Training cost comparison on 8×H20 GPUs. We report time per epoch and total time costs ("overall").

| Method | Books | | Movies | |
|---|---|---|---|---|
| | time/epoch | overall | time/epoch | overall |
| Exp3RT | 0.97h | 2.91h | 0.23h | 0.70h |
| SIREN–GRPO | 9.18h | 18.35h | 3.80h | 7.61h |
| SIREN–HA | 0.39h | 1.18h | 0.14h | 0.42h |
| **SIREN (total)** | — | **19.53h** | — | **8.03h** |

advantage for the $i$-th response is:

$$\hat{A}_{i,t} = \frac{R_i - \text{mean}(\{R_j\}_{j=1}^G)}{\text{std}(\{R_j\}_{j=1}^G)}. \tag{13}$$

The policy is then optimized using a clipped objective similar to PPO, with an additional KL regularization term to control deviation from the reference policy $\pi_{\text{ref}}$:

$$\mathcal{J}_{\text{GRPO}}(\theta) = \mathbb{E}\left[\frac{1}{G}\sum_{i=1}^G \frac{1}{|o_i|}\sum_{t=1}^{|o_i|} \min\left(r_{i,t}(\theta)\hat{A}_{i,t}, \; \text{clip}(r_{i,t}(\theta), 1-\varepsilon, 1+\varepsilon)\hat{A}_{i,t}\right) - \beta D_{\text{KL}}(\pi_\theta \,\|\, \pi_{\text{ref}})\right], \tag{14}$$

where $r_{i,t}(\theta)$ is the importance ratio between the current and old policies. GRPO thus leverages group-relative normalization to avoid reward hacking and removes the dependency on a value function.

In our setting, GRPO trains LLMs to produce explicit reasoning (CoT) for rating prediction, with a combination of format reward and rating regression reward as overall reward. This provides a foundation for our internalized reasoning stage, where we align hidden states to enable answer-only inference.

**Qualitative analysis of GRPO-trained reasoning over the interest sketch.** We examine how GRPO enables explicit CoT reasoning over the sketch prompt. Table 14 shows a Movies test case. The model (i) grounds its rationale in the likes/dislikes sketch (e.g., leveraging "military conflict & combatant perspectives"), (ii) cross-checks the candidate against recent interactions (aligning *Godzilla* with the user's high rating on *Rogue One*), and (iii) produces a clean `<think>`...`</think>` rationale followed by a numeric prediction between `<answer>`...`</answer>`. This illustrates that GRPO-trained CoT can make good leverage of interest sketch to filter noise, highlight stable preferences, and yield interpretable alignment between evidence and prediction.

## D  PROMPTS

Table 12 shows the CoT prompt template used for rating prediction during GRPO training (Sec. 3.2). Table 13 presents the prompt for summarizing cluster $k$ (constructed from the $M$ items nearest to $\mu_k$) into a concise topic name.

## E  LLM USAGE

We used a large language model solely for language polishing (grammar, phrasing, and copy-editing). It did not contribute to research ideation, experiment design, implementation, data analysis, or the creation of technical content (e.g., equations, algorithms, or results). All scientific claims, datasets, code, and citations were produced and verified by the authors, who take full responsibility for the content.

Table 12: CoT prompt template used for rating prediction during GRPO training.

**SYSTEM PROMPT**

You are an intelligent recommender system assistant. Your task is to predict the rating (from 1.0 to 5.0) that a user will give to a candidate `{category}`, based on the user's past behavior and preferences.

You should first reason about how well the candidate item aligns with the user's historical preferences, and then output a predicted rating. Your reasoning and the final rating must be wrapped with `<think> </think>` and `<answer> </answer>` tags, respectively.

The final rating must be a numeric value between 1.0 and 5.0. Do not include any extra explanation after the `<answer>` tag.

**ASSISTANT PROMPT TEMPLATE**

The user's long-term interest sketch is:
`{long-term user sketch}`
Below are the user's recent `{category}` ratings in the format: Title, Genres, Rating. Ratings range from 1.0 to 5.0.
`{N most recent interactions}`
The candidate `{category}` is described as: `{candidate item description}`
Based on the above information, what rating will the user give?

Table 13: Prompt used to summarize cluster into a concise topic name.

**SYSTEM PROMPT**

You are an expert taxonomy editor. Given a set of `{M} {category}` entries representing items nearest to the cluster center, produce a single concise topic name that best summarizes their shared theme.
[INSTRUCTIONS]
Analyze the implicit commonalities across the provided `{category}` entries (title, tags, brief description). Focus on high-level semantics, not surface details.
[CONSTRAINTS]
1. Output a single short phrase ($\leq$ 10 words).
2. Use concise, abstract language (no full sentences).
3. Do **not** list specific titles or examples.
4. **Output format:** `Theme:   <concise phrase>`

**ASSISTANT PROMPT TEMPLATE**

Nearest-to-center items:
`{M item descriptions}`

Table 14: Qualitative example on GRPO-trained reasoning over the interest sketch.

**Prompt**

**[SYSTEM]**

You are an intelligent recommender system assistant. Your task is to predict the rating (from 1.0 to 5.0) that a user will give to a candidate movie, based on the user's past behavior and preferences.

You should first reason about how well the candidate item aligns with the user's historical preferences, and then output a predicted rating. Your reasoning and the final rating must be wrapped with `<think> </think>` and `<answer> </answer>` tags, respectively. The final rating must be numeric in $[1.0, 5.0]$. Do not include any extra explanation after the `<answer>` tag.

**[USER: Sketch]**

*Likes (ratings $\geq 4$):* Military Conflict & Combatant Perspectives
*Dislikes (ratings $\leq 3$):* Crime-solving with ethical ambiguity; Secrets & emotional turmoil in relationships

**[USER: Recent ratings (Title, Genres, Rating)]**
```
Superman:  The Animated Series, Volume 3 (DC Comics Classic Collection),
Movies & TV|Today's Deals|Featured Deals & New Releases|The Big DVD &
Blu-ray Blowout, 2
Romancing the Stone (Special Edition), Movies & TV|Fully Loaded
DVDs|Special Editions, 3
ROGUE ONE: A STAR WARS STORY, Movies & TV|Science Fiction &
Fantasy|Science Fiction|Star Wars, 5
```
**[USER: Candidate item]**
```
Godzilla:  King of the Monsters Special Edition (DVD), Movies & TV|Fully
Loaded DVDs|Special Editions
```

**Model Output**

```
<think>
```
Okay, let's see. The user has specific preferences and dislikes. They like ==military conflict== and combatant perspectives, which means they probably ==enjoy action-packed, maybe war-related or heroic stories==. They dislike crime-solving with ethical issues and secrets in relationships, so maybe they don't like complex moral dilemmas or emotional drama.

Looking at their recent ratings: Superman: The Animated Series got a 2, which is a bit low. That's a superhero movie, so maybe they like some action but not the animated style? Romancing the Stone is a 3, which is a romantic thriller. They might not like that much. Rogue One, a Star Wars movie, got a 5, so they like sci-fi, especially from Star Wars. The candidate movie is Godzilla: King of the Monsters Special Edition. That's a sci-fi monster movie with action elements. Godzilla films often have a lot of ==action and are set in a post-war context, which might align with their preference for military conflict==. The user liked Rogue One, which is also a high-action sci-fi. The Special Edition DVD might be a plus since they liked the Special Edition of Romancing the Stone. Considering their dislike for crime-solving and emotional issues, ==Godzilla's more about action and less about complex relationships, so it should fit.== The rating is likely a 5.
```
</think>

<answer>5.0</answer>
```

Table 15: Hyper-parameter (HP) settings of our method on each dataset. Shared values are repeated for completeness.

| HP | Books | Movies |
|---|---|---|
| *Backbone & Encoders* | | |
| Backbone LLM | Qwen3-4B | Qwen3-4B |
| Text encoder | BGE-M3 | BGE-M3 |
| *Sketch & Prompt Construction* | | |
| Topics ($K$) | 20 | 20 |
| Topic naming ($M$) | 100 | 100 |
| Like/Dislike threshold ($\theta$) | 4 | 4 |
| Recent history length ($N$) | 30 | 10 |
| *GRPO Training (Explicit Reasoning)* | | |
| Framework | VERL | VERL |
| max_prompt_length | 2046 | 2046 |
| max_response_length | 512 | 512 |
| learning_rate | $1 \times 10^{-6}$ | $1 \times 10^{-6}$ |
| KL coefficient | 0.001 | 0.001 |
| micro_batch_size_per_GPU | 32 | 32 |
| rollouts | 8 | 8 |
| epochs | 2 | 2 |
| *Hidden Alignment* | | |
| Target modules | K, V | K, V |
| LoRA rank ($r$) | 8 | 8 |
| lora_alpha | 16 | 16 |
| lora_dropout | 0.05 | 0.05 |
| optimizer | AdamW | AdamW |
| learning_rate | $1 \times 10^{-4}$ | $1 \times 10^{-4}$ |
| schedule | constant | constant |
| warmup_steps | 100 | 100 |
| batch_size (global) | 64 | 64 |
| epochs | 3 | 3 |
| distributed setup | DeepSpeed ZeRO-2 | DeepSpeed ZeRO-2 |
| hardware | $8\times$ H20 GPUs | $8\times$ H20 GPUs |
| *Latency Evaluation (Inference)* | | |
| engine | vLLM | vLLM |
| hardware | single H20 | single H20 |
| batch size | 1 | 1 |
| max_response_length | 1024 | 1024 |

