# OpenReview forum: "Token-Efficient Long-Term Interest Sketching and Internalized Reasoning for LLM-based Recommendation"
_ICLR.cc/2026/Conference — ICLR 2026 Poster_

### Official Review · Reviewer_NNTh · 2025-10-22

**Soundness:** 2
**Presentation:** 3
**Contribution:** 3
**Rating:** 4
**Confidence:** 2

**Summary:**

The paper proposes SIREN, a framework to improve rating prediction with large language models (LLMs). It targets two deployment challenges: (1) long and noisy user histories that harm reasoning, and (2) high inference latency due to explicit CoT generation.

**Strengths:**

1. Clear practical motivation: identifies real deployment issues of long user histories and expensive CoT reasoning.
2. Elegant design: the “interest sketch + hidden-alignment” combination is conceptually simple yet effective.
3. Significant efficiency gains: achieves near-CoT accuracy with minimal latency.

**Weaknesses:**

1. Limited dataset diversity. Experiments are conducted only on two domains (Books and Movies) from the Amazon Reviews dataset. The generalizability to ranking, Top-N recommendation, multimodal, or more complex scenarios remains unclear.
2. Dependence on textual descriptions. The robustness of the encoding and clustering components should be further validated under conditions where item descriptions are scarce, noisy, or cross-lingual.
3. Cross-model generalization. The evaluation is limited to Qwen3-4B. It would strengthen the paper to include experiments on models with different architectures and scales (e.g., Llama, Gemma).
4. Insufficient discussion of related work. Several relevant studies are not discussed or cited, such as [1–2] on long-term user profiling and [3] on latent reasoning.


[1] Temporal User Profiling with LLMs: Balancing Short‑Term and Long‑Term Preferences

[2] HyMiRec: A Hybrid Multi‑interest Learning Framework for Long‑Term Multi‑interest Sequential Recommendation

[3] Reinforced Latent Reasoning for LLM‑based Recommendation

**Questions:**

See comments in Weaknesses.

---

> ### Author Response · Authors · 2025-11-25
>
> Thank you for your constructive feedback. We have addressed all your comments below.
>
> > **W1:** Limited dataset diversity. Experiments are conducted only on two domains (Books and Movies) from the Amazon Reviews dataset. The generalizability to ranking, Top-N recommendation, multimodal, or more complex scenarios remains unclear.
>
> **Response:** As suggested, to evaluate generalizability beyond rating prediction, we conducted a candidate reranking experiment on IMDB following Exp3RT[1]’s protocol. Specifically, we use LightGCN [2] to retrieve the top-20 candidates per user, and LLM-based models rerank these candidates by predicted relevance. We compare SIREN against Exp3RT, with results summarized in the table below. SIREN consistently outperforms Exp3RT across all metrics; for example, SIREN improves Recall@5 by 14.7% and achieves approximately 52× lower latency, demonstrating both higher efficacy and efficiency.
>
> Combined with our stronger rating-prediction results reported in the paper, these findings support SIREN’s effectiveness and generalizability on diverse datasets. We have included the reranking experiment in Table 9, Section 4.3 of the revised paper.
>
> |Method|Recall@3|Recall@5|nDCG@3|nDCG@5|Latency(s)|
> |-|:-:|:-:|:-:|:-:|:-:|
> |Exp3RT|0.0161|0.0578|0.0149|0.0413|1.56|
> |SIREN|**0.0256**|**0.0663**|**0.025**|**0.0426**|**0.03**|
>
> [1] Review-driven Personalized Preference Reasoning with Large Language Models for Recommendation, SIGIR'25.
>
> [2] Lightgcn: Simplifying and powering graph convolution network for recommendation, SIGIR'20.
>
>
>
> > **W2:** Dependence on textual descriptions. The robustness of the encoding and clustering components should be further validated under conditions where item descriptions are scarce, noisy, or cross-lingual.
>
> **Response:** As suggested, we conducted experiments to evaluate the robustness of encoding and clustering in long-term interest sketching under conditions of scarce item descriptions, using only brief metadata (titles and categories) instead of detailed descriptions. We fine-tuned the LLM with answer-only labels on these brief descriptions and compared its performance to the original SIREN implementation, which uses full descriptions. The results are shown below. We find that: (1) Brief descriptions yield slightly lower performance than detailed ones, confirming that richer textual information improves modeling; (2) The performance drop is modest, indicating that LLMs pretrained on large corpora can still extract meaningful semantics from limited text. Since detailed descriptions are available in the original datasets [1] and are straightforward to obtain, we recommend leveraging richer textual content for encoding and clustering to maximize robustness. These results are included in Table 6, Section 4.3 of the revised paper.
>
> |Methods|Books||Movies||
> |-|-:|-:|-:|-:|
> ||MAE(↓)|RMSE(↓)|MAE(↓)|RMSE(↓)|
> |Sketch(detail)|0.3536|0.7114|0.7695|1.3556|
> |Sketch(brief)|0.3594|0.7159|0.7706|1.3550|
>
> > **W3:** Cross-model generalization. The evaluation is limited to Qwen3-4B. It would strengthen the paper to include experiments on models with different architectures and scales (e.g., Llama, Gemma).
>
> **Response:** As suggested, we conducted backbone sensitivity experiments by evaluating SIREN on Llama-3.2-3B-Instruct, replacing Qwen3-4B while keeping all training settings identical. As shown in the table below, SIREN-CoT on Llama-3.2-3B achieves improved performance over Qwen3-4B on both datasets. The AO variant remains comparable across backbones and retains the 1-token latency advantage. On both architectures, SIREN (CoT and AO) consistently outperforms the Exp3RT baseline, demonstrating SIREN’s robustness and effectiveness across different model scales and architectures.
> We have included the experiment in Table 8, Section 4.3 of the revised paper.
>
>
> |Backbone|Method|Books||Movies||
> |-|-|-:|-:|-:|-:|
> |||MAE(↓)|RMSE(↓)|MAE(↓)|RMSE(↓)|
> |Qwen3-4B|Exp3RT|0.4001|0.7567|0.9521|1.4646|
> ||SIREN-CoT|0.3521|0.6962|0.7577|1.2822|
> ||SIREN-AO|0.3503|0.6887|0.7603|1.2924|
> |Llama-3.2-3B-Instruct|Exp3RT|0.3616|0.7048|0.9209|1.4157|
> ||SIREN-CoT|0.3481|0.6956|0.7524|1.2682|
> ||SIREN-AO|0.3534|0.7042|0.7622|1.2902|

---

> ### Author Response · Authors · 2025-11-25
>
> > **W4:** Insufficient discussion of related work. Several relevant studies are not discussed or cited, such as [1–2] on long-term user profiling and [3] on latent reasoning.
>
> **Response:** As suggested, we have added these works to the Related Work section. Specifically, for modeling user long-term interests, LLM-TUP [1] uses LLMs to generate short-term and long-term preference descriptions, which are encoded and fused via attention to construct semantically rich user embeddings. HyMiRec [2] reconstructs full historical sequences using residual codebook quantization and aggregates them into coarse interest embeddings with a lightweight recommender. These coarse interests are then concatenated with the last-n recent interactions as inputs to the LLM recommender for refined interest modeling. In contrast, SIREN performs dataset-level topic discovery once (via text encoding and clustering) and represents each user with a compact, fixed-budget, human-readable like/dislike topic list. Notably, LLM-TUP and HyMiRec process entire long sequences with Transformer-style models, incurring $O(N^2)$ attention complexity with respect to history length, whereas SIREN encodes each item independently and maps it to a topic cluster, yielding $O(N)$ complexity and stable token budgets.
>
> For LatentR3 [3], we have discussed it in the Inference Acceleration for LLM-based Recommendation section of Related Work in the original paper. To further clarify, LatentR3 introduces a latent reasoning module that generates a small number of autoregressive latent tokens instead of long text tokens, and trains them with a modified GRPO framework. In contrast, our internalized reasoning aligns hidden states so the model outputs the final answer directly, with no rationale tokens and no extra inference cost.
>
> [1] Temporal User Profiling with LLMs: Balancing Short‑Term and Long‑Term Preferences
> [2] HyMiRec: A Hybrid Multi‑interest Learning Framework for Long‑Term Multi‑interest Sequential Recommendation
> [3] Reinforced Latent Reasoning for LLM‑based Recommendation

---

> ### Comment · Reviewer_NNTh · 2025-11-27
>
> Thanks for the authors’ detailed response. I have carefully reviewed the rebuttal and the revised manuscript. I am pleased to see the addition of comprehensive experiments and a more thorough discussion of related work. These revisions significantly enhance the overall quality of the paper.
>
> Given that efficiency is essential in recommender systems, I find the proposed approach both meaningful and effective. I appreciate the contribution of this work and therefore decide to increase my score from 4 to 8.

---

> > ### Author Response · Authors · 2025-11-28
> > **Thank you for your effort and for increasing the score**
> >
> > Dear Reviewer NNTh,
> >
> > We appreciate your great efforts in having carefully reviewed our rebuttal and revised manuscript. We are pleased that our responses have addressed your comments, and we thank you for increasing the score.
> >
> > Best,
> > Authors

---

### Official Review · Reviewer_5SEj · 2025-10-31

**Soundness:** 4
**Presentation:** 4
**Contribution:** 3
**Rating:** 6
**Confidence:** 3

**Summary:**

This paper identifies two main limitations of LLM-based recommender systems: their difficulty in reasoning over long and noisy user histories, and their high inference latency. To address these challenges, the authors propose a method called SIREN, which employs long-term interest sketching to effectively process user histories and internalized reasoning to enhance inference efficiency. Experimental results demonstrate that the proposed method performs well.

**Strengths:**

1. The paper is well-motivated, and the two main limitations of LLM-based recommender systems it identifies are highly worth addressing.
2. Experimental results show that the proposed method effectively mitigates these limitations, demonstrating solid empirical performance.
3. The hidden state alignment component is particularly insightful.

**Weaknesses:**

1. In lines 254–256, the paper states that *“the final rating prediction under both answer-only (Fig. 1(c)) and CoT (Fig. 1(d)) decoding depends on the hidden state at the <answer> token.”* Could you provide additional evidence to support this claim?
2. What does the “Rank” column mean in Table 2?
3. Could you compare the training costs of these LLM-based recommender systems?
4. I think Table 2 is missing an important baseline — one where the **first stage** extends the recent history with additional past interactions until reaching the token budget (*More history*), and the **second stage** uses **GRPO-CoT**.

**Questions:**

see weakness.

---

> ### Author Response · Authors · 2025-11-25
>
> We appreciate your constructive feedback. Below, we have  addressed each of your comments.
>
> > **W1:** In lines 254–256, the paper states that “the final rating prediction under both answer-only (Fig. 1(c)) and CoT (Fig. 1(d)) decoding depends on the hidden state at the `<answer>` token.” Could you provide additional evidence to support this claim?
>
> **Response:** As suggested we provide additional analysis and reference to support this claim.  In an autoregressive decoder-only LLM, the next-token distribution is computed from the last hidden state at the output position via a linear projection and softmax:
> $p(y \mid x_{1:t}) = \mathrm{softmax} \big( W_{\mathrm{out}}\, h_t + b \big)$,
> where $h_t$ is s the hidden state at the `<answer>` position given the entire prefix (with or without CoT), and $W_{\mathrm{out}}$ is the output projection. The chosen rating token is
> $\hat{y}=\arg\max_{y}p \left(y \mid x_{1:t}\right).$ Thus, regardless of answer-only (AO) or CoT decoding, the final prediction depends only on the hidden state $h_t$ at `<answer>` (the prefix influences the result only through its effect on $h_t$). This is the standard Transformer LM decoding rule [1].
>
> [1] Attention Is All You Need. NIPS'17.
>
>
> > **W2:** What does the “Rank” column mean in Table 2?
>
> **Response:** The “Rank” column reports the average rank of each method across all datasets and both metrics (MAE, RMSE). For each metric on each dataset, methods are ranked (1 = best). The final “Rank” for a method is the mean of its ranks over all metrics and datasets, as shown in Table 2. A lower “Rank” indicates better overall performance. This explanation has been added to the Table 2 caption in the revision.
>
>
>
> > **W3:** Could you compare the training costs of these LLM-based recommender systems?
>
> **Response:** As suggested, we compare the training costs of SIREN and the LLM-based recommender Exp3RT under identical hardware (8×H20), with results summarized in the table below. Since SIREN employs two-stage training—GRPO followed by hidden alignment (Sec. 3.2)—we report the time for each stage as well as the total training time. SIREN requires more training time, mainly due to the GRPO stage, which is typically slower in LLM training because it involves multi-sample rollouts and autoregressive decoding overhead [1]. Nevertheless, SIREN achieves substantially lower online inference latency (see Fig. 3) and higher accuracy (see Table 2), making it a worthwhile trade-off between accuracy, online inference efficiency, and offline training costs. For future work, we plan to investigate cost reduction strategies such as coreset selection for smaller training sets [2] and dynamic sampling in RL [3] to improve efficiency. We have included the experiments in Table 11, Appendix B.4 of the revised paper.
>
> |Method|Books||Movies||
> |-|-:|-:|-:|-:|
> ||time/epoch|overall|time/epoch|overall|
> |Exp3RT|0.97h|2.91h|0.23h|0.70h|
> |SIREN-GRPO|9.18h|18.35h|3.80h|7.61h|
> |SIREN-HA|0.39h|1.18h|0.14h|0.42h|
> |SIREN||19.53h||8.03h|
>
> [1] FastGRPO: Accelerating Policy Optimization via Concurrency-aware Speculative Decoding and Online Draft Learning.
> [2] Importance-Aware Data Selection for Efficient LLM Instruction Tuning, AAAI'26.
> [3] DAPO: An Open-Source LLM Reinforcement Learning System at Scale.
>
> > **W4:** I think Table 2 is missing an important baseline — one where the first stage extends the recent history with additional past interactions until reaching the token budget (More history), and the second stage uses GRPO-CoT.
>
> **Response:** We have implemented the baseline as you suggested, where the input prompt extends the recent history by including additional past interactions until the token budget is reached, and the model is trained using GRPO-CoT with the same reward function and settings as SIREN. We compare its performance (More history–GRPO) with More history–SFT (using answer-only labels) and our proposed SIREN. The results are reported in the table below. We observe that (1) More history–GRPO, which generates explicit CoT before the final answer, generally outperforms More history–SFT, except for MAE on the Books dataset, demonstrating the benefit of reasoning. (2) SIREN consistently surpasses More history–GRPO, as it incorporates long-term interest sketching, which more effectively captures user preference signals under strict token limitations. We have included this baseline in Table 2, Section 4.2, and provide implementation details in Appendix B.3 of the revised paper.
>
> |Methods|Books||Movies||
> |-|-:|-:|-:|-:|
> ||MAE(↓)|RMSE(↓)|MAE(↓)|RMSE(↓)|
> |More history–SFT|0.3535|0.7153|0.7695|1.3723|
> |More history–GRPO|0.3587|0.7062|0.7640|1.3381|
> |SIREN|**0.3503**|**0.6887**|**0.7603**|**1.2924**|

---

### Official Review · Reviewer_Zz5G · 2025-10-31

**Soundness:** 2
**Presentation:** 3
**Contribution:** 2
**Rating:** 4
**Confidence:** 4

**Summary:**

The paper proposes a post-training method for LLM-based recommendation systems that can reduce input length by aggregating and clustering long histories. Through a two-stage training algorithm, it first provide the LLM with accurate recommendation capabilities, then uses hidden state supervision alignment to reduce the CoT part to zero, thereby achieving fast and accurate score computation. The work's motivation and experiments are relatively comprehensive, and the writing is fairly well-structured. However, the overall innovation is notably insufficient. In particular, the results of the ablation study (RQ4) actually demonstrate that the proposed method's optimizations for the input and output components show minimal improvement in effectiveness.

**Strengths:**

1. The paper proposes a two-stage post-training method based on reinforcement learning that first elicits the model's reasoning capability through RL to achieve accurate score estimation, then shortens the CoT length through an aligned "internalized reasoning" stage, thereby accomplishing the objective of being "both fast and accurate."

2. The motivation is clear and highly valuable, addressing a problem that current LLM-based systems are actively working to solve. The paper provides comprehensive experimental results demonstrating the superiority of the proposed method over other related approaches, and is also well-written.

**Weaknesses:**

1. The paper suffers from a notable lack of innovation, applying mature techniques from the existing community to recommendation systems. However, neither Stage 1 nor Stage 2 represents a novel idea. Moreover, the ablation study results (Figure 4) indicate that these modifications show almost no difference from direct SFT (CE in the Figure 4), with only marginal improvements (considering the test set and randomness, my personal view is that these improvements are extremely minimal).

2. As for the experimental results. Particularly in Figure 4, where simple CE achieves results very close to the original SIREN, I am unclear why your reproduced Exp3RT performs significantly worse (I noticed in the appendix that you used the same base model for training, but still have doubts about the results). Additionally, it should be noted that I did not find SFT results directly using ratings as labels in the paper, so I can only refer to your SFT results based on the Stage 1 model as the base (i.e., CE in Figure 4).

3. There are also some minor writing issues, such as subscripts that should be in roman type but were overlooked (e.g., Eq. 12), "Appendix ??" on line 491, and non-standard citation formatting (e.g., the title of Kim 2025's paper is incorrect)

**Questions:**

In addition to the issues mentioned in the Weaknesses section, I would also like to ask the authors:
1. The instability and randomness of RL are widely acknowledged in the community. Have the authors encountered similar issues, and were the experimental results averaged over multiple training runs as is common practice in the RL community?
2. When selecting the reward function, was an ablation study conducted (especially for s_{rate}) to try different forms of reward functions?

---

> ### Author Response · Authors · 2025-11-25
>
> Thank you for the constructive feedback. Below, we have addressed all comments.
>
> > **W1.1:** The paper suffers from a notable lack of innovation, applying mature techniques from the existing community to recommendation systems. However, neither Stage 1 nor Stage 2 represents a novel idea.
>
> **Response:** We respectfully argue that our proposed **long-term interest sketching** techniques (Sec. 3.1) and **internalized reasoning** technique (Sec. 3.2) are **novel**, different from existing methods.
>
> Specifically, the *long-term interest sketching* method comprehensively captures user preference signals under strict token limits, unlike prior LLM-based recommendation systems that either truncate to the most recent items—thus neglecting long-term interests—or require large LLMs to summarize long histories for each user, which still involves processing long contexts (a known weakness of LLMs [1]) and is computationally costly. To address this, we propose the **Item Topic Discovery** technique, which performs corpus-level topic detection once via lightweight text encoding and clustering, then maps each item to these topics, thereby avoiding per-user long-context summarization. Furthermore, the **User Interest Sketching** technique builds a token-budgeted, human-readable profile by aggregating a user’s interactions into a small, fixed-size like/dislike topic list that remains stable even with very long histories. Table 3 in the paper demonstrates the effectiveness of our proposed interest sketching compared to truncated histories and user summaries.
>
> The **internalized reasoning** method enables our model to directly output the final rating without sacrificing the quality gains of CoT. We introduce a two-stage training paradigm that first teaches the model to reason, then internalizes that reasoning. In Stage 1, we train explicit CoT with GRPO and our proposed rating-regression reward $s_{\mathrm{rate}}$ instead of an exact match reward. $s_{\mathrm{rate}}$ provides a denser learning signal by rewarding near-correct predictions rather than all-or-nothing feedback, which helps further stabilize GRPO updates. In Stage 2, our proposed hidden alignment enables the model to directly output the answer—**no intermediate tokens at all, no extra costs at inference**. In contrast, efficient reasoning strategies like latent reasoning or speculative decoding still produce or verify intermediate tokens and incur extra inference cost. Table 5 shows that internalized reasoning achieves approximately *100× faster latency* than CoT at comparable accuracy, meeting real-world recsys latency requirements.
>
> We have included this clarification in the Related Works, Section 5, of the revised paper.
>
> [1] Context Length Alone Hurts LLM Performance Despite Perfect Retrieval, EMNLP'25.

---

> ### Author Response · Authors · 2025-11-25
>
> > **W1.2:** Moreover, the ablation study results (Figure 4) indicate that these modifications show almost no difference from direct SFT (CE in the Figure 4), with only marginal improvements (considering the test set and randomness, my personal view is that these improvements are extremely minimal).
>
> > **W2:** As for the experimental results. Particularly in Figure 4, where simple CE achieves results very close to the original SIREN, I am unclear why your reproduced Exp3RT performs significantly worse (I noticed in the appendix that you used the same base model for training, but still have doubts about the results). Additionally, it should be noted that I did not find SFT results directly using ratings as labels in the paper, so I can only refer to your SFT results based on the Stage 1 model as the base (i.e., CE in Figure 4).
>
> **Response:** We address W1.2 and W2 together, as they are closely related.
>
> First, we clarify that Fig. 4 in the paper only compares different strategies for internalizing reasoning. In other words, all variants in Fig. 4 still utilize our proposed SIREN framework, which learns explicit CoT reasoning over the long-term interest sketch, trained with GRPO and our rating regression reward (Sec. 3.2). The only difference among these variants is their internalized reasoning strategy. Thus, CE in Fig. 4 is **not a direct SFT ablation**.
>
> To address this comment on ablation performance, we conducted new ablation experiments and report the results for direct (from-scratch) SFT alongside our two-stage internalized reasoning paradigm (Sec. 3.2) below. Specifically, using the same long-term interest sketch as input, we compare:
> (a) **Direct SFT**: trained from scratch on rating labels using the sketch; inference is answer-only.
> (b) SIREN Stage 1: model trained to explicitly reason using GRPO with our proposed rating regression reward; inference with CoT.
> (c ) SIREN Stage 2: starting from Stage 1 (b), model trained to internalize reasoning via hidden alignment; inference is answer-only.
>
> In the table below, comparing (b) to (a), MSE and RMSE consistently decrease (notably in RMSE; e.g., (b) reduces RMSE by 1.9% on Books and 5.4% on Movies). (c ) achieves a dramatic reduction in inference latency compared to (b), while maintaining comparable accuracy, thus balancing efficiency and efficacy. Overall, these results validate the effectiveness of our proposed techniques in Section 3 for SIREN, and the improvements are systematic rather than random.
>
> We have clarified the Fig. 4 setup and integrated this ablation in Table 5, Section 4.3 of the revision.
>
> |Method|**Books**|||**Movies**|||
> |-|-:|-:|-:|-:|-:|-:|
> ||MAE(↓)|RMSE(↓)|Latency(↓)|MAE(↓)|RMSE|Latency(↓)|
> |(a) Direct SFT|0.3536|0.7114|0.013|0.7695|1.3556|0.013|
> |(b) SIREN Stage 1|0.3521|0.6977|2.22|0.7577|1.2821|2.15|
> |(c ) SIREN Stage 2|0.3503|0.6887|0.013|0.7603|1.29242|0.013|
>
> Regarding Exp3RT, we used the authors’ code and the Books dataset from their repository. Exp3RT’s generated profile includes average user/item ratings, which strongly affect outcomes. However, we found that the averages given by their release do not match those recomputed from the training split only (the latter avoids leakage). For a fair comparison, we recomputed these averages on the training data only and report the corrected results (see Appendix B.3). We have further clarified all Exp3RT implementation details in Appendix B.3 of the revised paper.
>
> > **W3:** There are also some minor writing issues, such as subscripts that should be in roman type but were overlooked (e.g., Eq. 12), "Appendix ??" on line 491, and non-standard citation formatting (e.g., the title of Kim 2025's paper is incorrect)
>
> **Response:** As suggested, we have addressed the minor writing issues in the revision. Specifically, subscripts now use roman type where appropriate (e.g., Eq. 12), the Appendix cross-reference has been resolved, and all citations have been standardized, including the correct title for Kim 2025.

---

> ### Author Response · Authors · 2025-11-25
>
> > **Q1:** The instability and randomness of RL are widely acknowledged in the community. Have the authors encountered similar issues, and were the experimental results averaged over multiple training runs as is common practice in the RL community?
>
> **Response:** We clarify that all experiments are conducted with three random seeds, and we report the mean performance, consistent with prior work such as Exp3RT. This setting is clarified in the experimental details, Section 4.1 of the revised paper.
>
> Regarding RL stability, modern RL approaches for LLMs such as GRPO employ techniques including KL penalty to a reference model and reward clipping, which substantially enhance training robustness [1]. Additionally, our rating-regression reward $s_{\mathrm{rate}}$ (Sec. 3.2) provides a denser learning signal by rewarding near-correct predictions rather than all-or-nothing feedback, further stabilizing GRPO updates. With these measures, our RL training remained stable and did not exhibit large oscillations.
>
> [1] Revisiting Group Relative Policy Optimization: Insights into On-Policy and Off-Policy Training
>
> > **Q2:** When selecting the reward function, was an ablation study conducted (especially for s_{rate}) to try different forms of reward functions?
>
> **Response:** As suggested, we conducted an ablation study comparing our proposed linear rating-regression reward $s_{\mathrm{rate}}$ (Eq. 11) with a binary exact-match reward $s_{\mathrm{binary}}$ that assigns $+2$ when the predicted rating exactly matches the ground-truth rating and $-2$ otherwise. The format reward and all other GRPO settings are kept unchanged. The results are summarized in the table below.
>
> Observe in the table below that SIREN trained with $s_{\mathrm{rate}}$ consistently outperforms training with $s_{\mathrm{binary}}$. We attribute this to two main factors: (1) $s_{\mathrm{rate}}$ provides a denser learning signal by rewarding near-correct predictions, while $s_{\mathrm{binary}}$ offers only all-or-nothing feedback; (2) $s_{\mathrm{rate}}$ yields smoother reward changes with error, resulting in more stable GRPO training.
>
> This ablation study has been added to Table 7, Section 4.3 of the revised paper.
>
>
> |Methods|Books||Movies||
> |-|-:|-:|-:|-:|
> ||MAE(↓)|RMSE(↓)|MAE(↓)|RMSE(↓)|
> |$s_{\mathrm{binary}}$|0.3555|0.6997|0.8094|1.4201|
> |$s_{\mathrm{rate}}$|**0.3520**|**0.6962**|**0.7577**|**1.2822**|

---

### Author Response · Authors · 2025-12-03
**Rebuttal Summary to AC**

Dear ACs,

We have addressed all reviewer comments. As a result, the rating has increased from 4/6/4 to **8/6/4**. We thank the reviewers and ACs for their thoughtful feedback and efforts.

Below, we summarize our key contributions and the revisions made in the rebuttal.

Our key contributions are as follows (with reviewer quotes):

* **Novel Method for a Well-motivated Problem.** We address two core limitations of LLM-based recommender systems: difficulty in reasoning over long and noisy user histories, and high inference latency. We propose SIREN, a novel method that couples token-efficient long-term interest sketching with internalized reasoning to meet real-world deployment requirements. As noted by reviewers, the motivation is *clear and highly valuable*, directly addressing a problem that current LLM-based systems are actively working to solve (Zz5G, 5SEj, NNTh). To overcome these limitations, our method yields *elegant and insightful designs* (NNTh, 5SEj), and comprehensive experimental results demonstrate the *superiority of the proposed approach* (Zz5G, 5SEj, NNTh).


* **Effective and Efficient Technical Designs.** We introduce key technical components that realize the SIREN framework, recognized by reviewers as an elegant “interest sketch + hidden-alignment” combination—conceptually effective (NNTh), insightful (5SEj), and achieving both speed and accuracy (Zz5G):

    * **Effective token-efficient user modeling.** Our *Long-Term Interest Sketching* leverages dataset-level *Item Topic Discovery* and per-user *User Interest Sketching* to compress full user histories into compact, human-readable, token-budgeted profiles. This approach avoids per-user long-context summarization while preserving stable long-term signals.
    * **Efficient internalized reasoning for 1-token inference.** We propose a two-stage training paradigm that enables the model to output the final prediction in *a single token*. Specifically, (i) explicit reasoning for rating prediction is learned over long-term interest sketch prompts *without CoT labels via RL*, and (ii) this reasoning is internalized through *hidden-state alignment*, allowing answer-only decoding with near-CoT quality. Unlike latent-reasoning or speculative-decoding methods, our approach generates no intermediate tokens and incurs *no extra inference cost*.
* **Comprehensive Experiments** We provide extensive experimental evidence demonstrating the superiority of our proposed method over related approaches (Zz5G) and its effectiveness in addressing existing limitations, with strong empirical performance (5SEj) and substantial efficiency gains: near-CoT accuracy with minimal latency (NNTh). SIREN consistently outperforms strong LLM-based baselines (e.g., Exp3RT) in both accuracy and inference latency, generalizes across model backbones (Qwen3-4B and Llama-3.2-3B-Instruct), and remains robust under constrained item descriptions as well as on an additional IMDB reranking task. SIREN achieves CoT-level accuracy with up to ~100× lower latency.


In the rebuttal, we have addressed all reviewer concerns and incorporated the following major revisions:

* For **Reviewer 5SEj**, we improved technical clarity and empirical rigor by:
    * Providing a more explicit explanation of why the final rating prediction in both answer-only and CoT decoding depends on the hidden state at the <answer> token. (W1)
    * Clarifying the meaning of the “Rank” column in Table 2. (W2)
    * Adding a detailed training cost comparison between SIREN and other LLM-based models. (W3)
    * Implementing and reporting the “More history–GRPO” baseline. (W4)

* For **Reviewer NNTh**, we broadened the evaluation scope and expanded the related-work discussion, fully addressing the reviewer’s concerns and resulting in an **updated score from 4 to 8**:
    * Added an IMDB reranking experiment, showing SIREN outperforms Exp3RT in accuracy and achieves ~52× lower latency, demonstrating effectiveness beyond rating prediction. (W1)
    * Validated the robustness of long-term interest sketching under limited textual descriptions (titles + categories only). (W2)
    * Conducted backbone sensitivity experiments on Llama-3.2-3B-Instruct, confirming SIREN’s cross-architecture generalization. (W3)
    * Expanded the Related Work section to cover additional long-term profiling and latent reasoning methods. (W4)

* For **Reviewer Zz5G**, we clarified the novelty of SIREN (W1) and added new ablation studies:
    * A direct-from-scratch SFT baseline. (W2)
    * An ablation of the reward function, demonstrating $r_{rate}$ yields consistently better and more stable GRPO training. (Q2)
    * Additional clarifications on Exp3RT reproduction (W2), experimental protocol (Q1), RL stability under GRPO (Q1), and thorough correction of writing issues (W3).


Thank you.


Best,

Authors

---

### Meta-Review · Area_Chair_9hVN · 2026-01-06

**Summary:**

In this paper, the authors study the application of LLMs in recommendation systems, and try to address two of its practical challenges: 1) dealing with long user histories of interacting with different items (movies, books, etc.), and 2) the inference-time latency due to the need for generating long CoTs (LLMs with CoT have better performance). They propose SIREN, a LLM-based rating predictor. They first build a compact representation of the user's history as a short list of liked/disliked topics, which is then used to prompt the LLM in place of the long user's history. Then, they use a two-stage training strategy where they first train the LLM for rate prediction using RL, followed by learning to internalize CoT into model parameters through hidden alignment, such that at inference time LLM generates the rating with near CoT quality.

- Most reviewers believe that the paper is well-motivated and clearly written. Moreover, they found the two main ingredients of the work, namely compact representation of user's history (interest sketch) and internalizing CoT through hidden-alignment, despite their simplicity, highly effective. Finally, they believe the main claims of the paper are properly supported by the experiments.

- Two of the main concerns of the reviewers, insufficient discussion of related work and some issues related to the experiments, were addressed reasonably well by the authors during the rebuttals. Although, I still believe the paper can benefit from more discussion on the related work and its quality can improve by better positioning it within the literature of using LLMs in RecSys.

- I agree with some reviewers that the main ingredients of the work are simple but I think they have been put together nicely to address some of the main limitations of using LLMs in recommendation systems.

**Reviewer Concerns:**

It seems the authors successfully addressed the issues raised by Reviewer NNTh, thus, they changed their score from 4 to 8.

**Reviewer Scores:**

Two of the reviewers did not participate in discussion with the authors. However, it seems the main concerns of Reviewer NNTh were addressed by the authors through more experiments and more discussion on related work, thus, the reviewer raised their score from 4 to 8.

---

### Decision · Program_Chairs · 2026-01-26

Accept (Poster)